# Autonomous Motivation as a Mediator Between an Empowering Climate and Enjoyment in Male Volleyball Players

**DOI:** 10.3390/sports7060153

**Published:** 2019-06-25

**Authors:** Sofia Mosqueda, Jeanette M. López-Walle, Pablo Gutiérrez-García, Juan García-Verazaluce, José Tristán

**Affiliations:** 1Facultad de Organización Deportiva, Universidad Autónoma de Nuevo León, San Nicolás de los Garza 66455, Mexico; smo94g@hotmail.com (S.M.); juan011@hotmail.com (J.G.-V.); 2Licenciatura en Entrenamiento Deportivo, Universidad Estatal de Sonora, Hermosillo 83170, Mexico; pablyabefe@hotmail.com

**Keywords:** motivational climate, volleyball players, well-being

## Abstract

The objective of this work was to analyze a mediation model concerning the perception of an empowering climate generated by a coach and enjoyment through the autonomous motivation of athletes. The sample consisted of 71 elite male volleyball players from six countries. The age range was 14 to 18 years (M = 16.5, SD = 0.96). The relationships between the perception of an empowering climate, autonomous motivation, and enjoyment were positive and significant. The mediation model showed that autonomous motivation acts as a mediator in the relationship between the perception of an empowering climate generated by the coach and the enjoyment reported by the athletes.

## 1. Introduction

One of the main objectives of the current psychological research is to discover mechanisms that promote health to improve the quality of life of individuals, and one of the main indicators of optimal functioning is psychological well-being, which seeks the development of capacities and personal growth. However, this concept is presented as a multifaceted and dynamic construct that includes subjective, social, and psychological components. Psychological well-being is perceived as a pleasant state, related to the equilibrium of our organism, the handling of emotions, and the control of our environment. It has its origin in ‘positive psychology’; this area integrates the positive outlook and optimistic aspects of health, such as the concepts associated with happiness and enjoyment [1]. Given this, it is to be assumed that those processes that help us to understand the relationship established between the individual and his environment, such as motivational and emotional processes, are strongly linked to psychological well-being [2].

Psychological well-being searches for the development of capacities and personal growth and is presented as a multifaceted and dynamic producer that includes subjective, social, and psychological components. This is perceived as a pleasant state, related to the balance of our organisms, the handling of emotions, and the control of our environment [2].

Psychological well-being has been studied by scientists from two differentiated perspectives; hedonic or subjective well-being, which seeks the maximization of pleasure, the presence of happiness, satisfaction with life, the minimization of pain, and eudaimonic or psychological well-being, which means that the well-being is determined by the accomplishment of activities that facilitate self-realization, in other words the individual looks for a virtuous life or one of excellence, where he feels capable and faces, with success, the proposed challenges [3].

Enjoyment is a relevant factor when seeking psychological well-being, because enjoyment is a positive emotion resulting both from satisfying a basic biological need and from successfully applying one’s own abilities to face perceived environmental challenges, thus orienting oneself towards growth [4].

Motivation is an important element in achieving commitment and adherence to sport, since it guides and determines the behavior and actions of the subject [5]. The self-determination theory (SDT) [6] indicates that human motivation develops in a continuum characterized by different levels of self-determination, considering that a self-determined behavior is autonomous, self-regulated, and based on psychological development and self-realization. Considering this, different types of motivation are exhibited, from higher to lower self-determination, and there is intrinsic motivation, extrinsic motivation, and amotivation [7]. Each of these types of motivation have a specific type of regulation, as well as a locus of causality.

The intrinsically motivated subjects present an autonomous orientation and interpret the events in an informative way. The behavior is self-determined and derives from an integrated sense of self, using the information to make choices and self-regulate towards chosen goals. On the other hand, individuals who present extrinsic motivation have behaviors carried out only to arrive at an end, that is to say, the events are made by pressure and not with a sense of one’s own election. Finally, those individuals with a lack of motivation present an erratic and unintentional functioning, considering the belief that the results are independent of behavior. In addition, the subject experiences a sense of lack of purpose. 

The SDT, as well as the achievement goal theory (AGT) [8], states that the social environment or motivational climate, created by other significant people, directly affects the motivation that the individual expresses [9]. As a result of these two theories, Duda [10] developed a hierarchical and multidimensional conceptualization of the motivational climate, claiming that it can be more or less empowering (task-involving, autonomy support, socially supportive) or disempowering (ego-involving and controlling style). Following the model of Duda, Appleton, Stebbings, and Balaguer [11], empowering and disempowering climates can influence the motivation of athletes and the quality of their sporting commitment, as well as facilitating or hindering sustained participation in sport. 

Empowering or disempowering climates have been associated with motivational regulations [9,12,13], autonomous motivation [12,14], self-determined motivation [9], and well-being indicators, such as enjoyment [11,12,14,15,16].

SDT and AGT are fundamental to determine psychological well-being, such as enjoyment. This is associated with the adherence to sport in young athletes [17].

Specifically, concerning the interaction of the three variables (empowering, motivation, and enjoyment), Fenton et al. [12] and Gutiérrez et al. [14] have found that perceptions of an empowering motivational climate has a positive relationship with players’ autonomous motivation and, in turn, sport-related enjoyment. Fenton et al. [12] have demonstrated that perceptions of an empowering climate have a significant positive indirect effect on enjoyment through autonomous motivation. Gutiérrez–García et al. [14] have demonstrated the moderating role of autonomous motivation between an empowering climate and enjoyment. Specifically, for high values of autonomous motivation, the relationship between an empowering climate and enjoyment was null, becoming a positive and significant relationship for low values of autonomous motivation.

Most studies use samples of young athletes, with school performance levels, so the aim of this study is to test the mediating role of autonomous motivation between the perception of the empowering climate generated by a coach and the enjoyment of elite athletes. We hypothesized that: (1) The perception of the empowering climate generated by the coach is positively related to the autonomous motivation and the enjoyment of the players; (2) the autonomous motivation is positively related to enjoyment in the players.

## 2. Materials and Methods

### 2.1. Measures

To measure the empowering climate, we used the Empowering and Disempowering Motivational Climate Questionnaire-Coach (EDMCQ-C) [18], used in the Mexican context [9,14]. The questionnaire has 32 items with two subscale, their a Cronbach internal reliability coefficients were satisfactory (α = .84–.96) [18], divided into five first-order subscales, and, for the purpose of this work, only 3 subscales were used, which were: Task-involved, which is formed by 9 items (e.g., “My coach encouraged players to try new skills”), autonomy support, with 5 items (e.g., “My coach gave players choices and options”), and social support, formed by 3 items (e.g., “My coach really appreciated players as people, not just as athletes”). The three first-order subscales are averaged to generate a second-order dimension, called the empowering climate. The answers are collected on a five-point Likert scale that ranges from “Totally disagree” (1) to “Totally agree” (5). 

To evaluate the autonomous motivation, we used a version of the sport motivation scale (SMS-II) [19], used in the Mexican context [20]. This questionnaire was composed of 18 items that were divided into six first-order subscales, each with three items that measured the motivational regulations. The Cronbach internal reliability coefficients of all subscales were satisfactory (α = .74–.83) [19]. The questionnaire began with the question, “Why do you participate in your sport?” and, for the purposes of this work, only 3 subscales were used, and these were: Identified regulation (e.g., “Because I have chosen this sport as a way to develop myself”), integrated regulation (e.g., “Because practicing sports reflects the essence of whom I am”), and intrinsic motivation (e.g., “Because it is very interesting to learn how I can improve”). To determine autonomous motivation, the values of these subscales were averaged. The answers were collected on a seven-point Likert scale, ranging from “Totally disagree” (1) to “Totally agree” (7). 

For the evaluation of enjoyment, we used a version of the sport satisfaction instrument (SSI) [21], used in the Mexican context [14]. Made up of 7 items with two subscales, their a Cronbach internal reliability coefficients were satisfactory (α = .75–.89) [14]., this instrument began with the introductory phrase, “In my sport...”, and was divided into 2 subscales: Enjoyment, with 5 items (e.g., “I normally have a good time practicing volleyball”) and boredom, with 2 items (e.g., “When I practice sport, I want the game to end quickly”). The answers were collected on a five-point Likert scale, ranging from “Completely disagree” (1) to “Completely agree” (5). 

The questionnaires were adapted to the context of volleyball, based on the versions used in Mexico.

### 2.2. Procedure

To carry out the data collection, we first contacted the International Volleyball Federation (FIVB) to request authorization for the application of the questionnaires within the competition “PanAmCup Boys Youth U19”, 18 to 26 March 2017, in Monterrey, Nuevo Leon, Mexico. Once we had this authorization, the leaders of the different participating teams were informed of the objective for the study and requested authorization for the application of the questionnaires to the athletes. The athletes signed a consent to participate. The time to answer the questionnaires was 30 min. The anonymity of the data collected was guaranteed throughout the process and it was made clear that the information would be used only for scientific purposes. The study has been approved and registered (REPRIN-FOD-29) by the Research Coordination of the School of Sports Organization.

### 2.3. Data Analyses

The IBM SPSS STATISTICS 24 software was used to analyze the data. First, an analysis of the items that made up the scales and subscales of the different questionnaires was carried out using descriptive statistics of the central tendency (mean), dispersion (standard deviation, minimum, and maximum values), and correlations through Pearson’s coefficient. Cohen’s criteria were used for the interpretation of each correlation as follows: Above 0.50 as large, between 0.30 and 0.50 as moderate, and between 0.10 and 0.30 as small [22]. We continued with the reliability analysis, the internal consistency through Cronbach’s Alpha coefficient, coefficient or reliability composite (RC) and the Spearman’s rank correlation coefficient (rho, ρ). Coefficient values equal to the greater reliability composite .70 indicate adequate reliability [23]. The mediation model was estimated by regression analysis, using the PROCESS macro for SPSS [24]. To test the indirect effect, bootstrap-corrected bias confidence intervals, with 5000 replications and a 95% confidence level, were used. This method involved calculating the product of the regression coefficients that estimated the indirect effect and obtained a confidence interval for that effect. If the confidence interval did not include the 0 value, the mediation effect was confirmed [24]. 

## 3. Results

### 3.1. Description of Participants

The participants were 71 male high-performance volleyball players, aged between 14 and 18 years old (M = 16.5, SD = 0.96), from six different countries (Mexico = 12, Puerto Rico = 12, Dominican Republic = 11, Nicaragua = 12, Jamaica = 12, and Peru = 12), who participated in the Pan American Cup Boys Youth U19, conducted in Monterrey, Nuevo Leon, Mexico from 16 to 26 March 2017. The athletes had an average of four years of practicing and competing in the sport and practiced for an average of four days a week (SD = 1.70), each daily session averaging 3.25 h (SD = 1.44). 

### 3.2. Descriptions, Reliability, and Correlations

The subscales of the questionnaires, used to evaluate the variables of the study, presented a measured stability using Cronbach’s Alpha, with values ranging from .65 to .96, which are considered acceptable by some authors [25,26], and the composite reliability above .70 [23]. Table 1 shows in greater detail the values of Cronbach’s Alpha, as well as the mean, standard deviation, minimum, and maximum of each variable. 

A Spearman’s rank correlation analysis was performed, where positive and significant relationships can be observed between the variables of autonomous motivation and empowering climate (.33), enjoyment and empowering climate (.36), and autonomous motivation and enjoyment (.59).

### 3.3. Mediation Model

In this model, we analyzed the mediating effect of autonomous motivation on the relationship between empowering climate and enjoyment, showing a significance of the .01 in the model, thus confirming that the perception of the empowering climate generated by the coach is positively related to the autonomous motivation and the enjoyment of the players. In addition, autonomous motivation is positively related to enjoyment in the players. The mediating role of autonomous motivation in the relationship between an empowering climate and enjoyment was confirmed (Figure 1). The results provide evidence of a total mediation effect.

## 4. Discussion

The main objective of this study was to test the mediator role of autonomous motivation in the relationship between the perception of the empowering climate generated by a coach and the enjoyment of elite athletes. In accordance with the postulates of the SDT, the AGT, and the Duda Model [11], the findings of this study show support for the hypothesized, considering that the environments generated by the significant agents, in this case, the coach, influences both the motivation and the enjoyment of the athlete who trains. In addition, the autonomous motivation is characterized by the search for enjoyment during practice. Therefore, enjoyment is an emerging component that aids adherence to the activity, creates a commitment to practice, develops a positive attitude, and helps to improve intra-group social interactions [27]. In this sense, it is concluded that more self-determined motivational profiles, namely, autonomous motivation, are related to greater enjoyment in exercise and therefore greater psychological well-being [28].

Confirming the mediating role of autonomous motivation in the relationship between empowering climate and enjoyment, it can be said that, if the athlete perceives that empowering climate has a high probability of developing an autonomous motivation, he/she therefore experiences enjoyment. This statement is in line with various studies [29,30,31].

The motivational climate generated by the coach is going to influence the behaviors, thoughts, and feelings that the athlete presents [11]. Therefore, determining whether to exhibit an empowering or disempowering climate within the sports training becomes a fundamental task when looking for high performance or persistence in the sport [32]. These motivational climates will condition the way in which athletes live their experience and, consequently, they will exhibit a type of motivation, which is a key element in determining whether the athlete presents a pro-active and committed attitude or, alternatively, a passive and inconsistent attitude [33].

If the athlete shows an intrinsic motivation, he will be expressing that motivation with greater self-determination, and this is characterized by the search for enjoyment during the practice. Therefore, as a result of its manifestation and the more self-determined regulations of extrinsic motivation, enjoyment is an emerging component that helps adherence and commitment to the activity, developing a positive attitude and auditioning to improve social interactions [27].

The variables that affect the enjoyment and the psychological well-being of athletes become vital for instructing coaches on how to promote the desired behaviors to achieve the optimal development of athletes.

These results highlight the importance of promoting an empowering climate in the context of high-performance sports, because it helps to develop autonomous motivation in athletes and favors the optimal development of enjoyment. Approaching it from a practical point of view, this study emphasizes the importance of coaches, developing an empowering climate for achieving optimal performance and positive development in their athletes. 

These results also tell us that, if coaches are instructed to generated an empowering climate within their training, giving them tools to promote a task-involving climate, promote the effort and accept error as a natural learning process in their athletes. In addition to fostering the autonomy of athletes by giving them decision-making power within their own training and encouraging socialization, helps their athletes to develop an autonomous motivation, in which the athlete seeks their own growth and development, thus having a better adherence to sports practice and their training, both visible (training sessions) and invisible (nutrition, rest etc.), and thus increasing their own sports performance. 

At the same time, the empowering climate will promote the enjoyment of the athlete, and if an autonomous motivation is attached, the enjoyment will be enhanced, which means that the athlete will have a positive affective response to the sports experience that reflects generalized feelings, such as pleasure, taste, fun, and psychological well-being, which is fundamental to achieve a state of health. In conclusion, this study emphasizes the importance of coaches developing an empowering climate for achieving optimal performance and positive development in their athletes. 

One of the limitations of this study was the fact that the sample was only of males and of a single sport, but it must be considered that it is dealing with high-performance athletes. Another limitation of this study is that the information is obtained only by subjective means derived from the perception of the athlete himself, so it would be interesting in future studies to include objective measures. Additionally, for future studies, it would be interesting to take this model to populations of both genders and to different sports, in addition to investigating the relationship of these variables directly in the physical performance of an objective form. 

With this study, it can be concluded that motivational empowering climates are related to enjoyment and self-motivation, so when an athlete perceives a motivational empowering climate, his enjoyment and self-motivation will be significantly increased. In addition, the mediating effect of self-motivation on the relationship between empowering climate and enjoyment was confirmed. 

## Figures and Tables

**Figure 1 sports-07-00153-f001:**
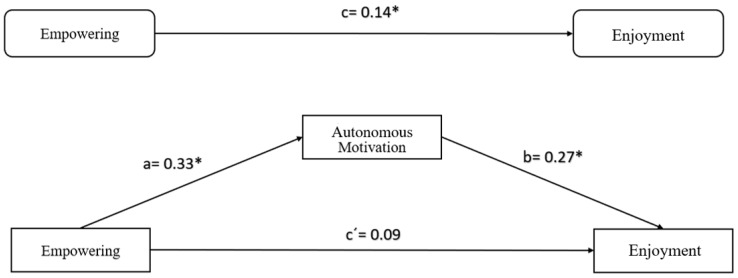
Model of mediation, empowering climate, autonomous motivation, and enjoyment. ** *p* < .01, * *p* < .05.

**Table 1 sports-07-00153-t001:** Descriptive, reliability, and correlations (ρ) between psychological variables.

	*M*	*SD*	Min	Max	Cronbach’s Alpha	RC	Empowering Climate	Autonomous Motivation	Enjoyment
Empowering Climate	4.47	.74	0.00	5	.96	.94	1		
Autonomous Motivation	6.27	.75	3.67	7	.85	.89	.33 **	1	
Enjoyment	4.75	.38	33	5	.65	.95	.36 **	.59 **	1

Reliability composite (RC), ** *p* < .01, * *p* < .05.

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
