# Peer review of "Autonomous Motivation as a Mediator Between an Empowering Climate and Enjoyment in Male Volleyball Players"

_sports, 2019, doi:10.3390/sports7060153_

Reviewer 1 Report

This paper presents a mediation analysis regarding the relationship between empowerment and enjoyment among high-performance youth volleyball players. Strengths of they study include good conceptualization of the variables and a strong analytic approach. I have only a few comments regarding the draft manuscript.

The methods section should include more details about the procedures for participant recruitment, and the methods used to administer the questionnaires.

Descriptive statistics regarding sample composition are currently presented in the Methods section (2.1), but this would belong better at the beginning of the Results section.

The connection between enjoyment of sports participation and psychological well-being is made in the introduction and implicitly in the discussion. While it makes a great deal of sense that enjoying one's activities would contribute to well-being, the logic of the paper would be strengthened by taking a sentence or two to make this connection explicit (preferably with some citations, if available).

Author Response

Response to Reviewer 1 Comments

Point 1: The methods section should include more details about the procedures for participant recruitment, and the methods used to administer the questionnaires.

Response 1: Added section 2.2 (Lines 130-139)

Point 2: Descriptive statistics regarding sample composition are currently presented in the Methods section (2.1), but this would belong better at the beginning of the Results section.

Response 2: Descriptive statistics regarding sample composition presented in the Methods section, was moved from the Results section (Lines 154-162)

Point 3: The connection between enjoyment of sports participation and psychological well-being is made in the introduction and implicitly in the discussion. While it makes a great deal of sense that enjoying one's activities would contribute to well-being, the logic of the paper would be strengthened by taking a sentence or two to make this connection explicit (preferably with some citations, if available).

Response 3: Were added paragraphs in the introduction to better describe this relationship, in addition, with some cites (lines 29-30, 34-47 and 79-82)

Reviewer 2 Report

My main concern refers to the presentation of the study and the explication of the method:

1)      Given the cross-sectional design, authors should avoid casual language. They should avoid the use of expression such as “predictort”, etc. given the study design authors should only talk about associations or relationships.  

2)      Introduction: I am not familiarize with the topic but I appreacite that authors built a strong theoretical framework to justify their study.

3)      Introduction should be end with concrete hypothesis to be test. Hypothesis should be drive in previous studies. I would appreciate an inclusion of a rationale for each hypothesis tested.

4)      Were all the selected athletes males or females? A rationale for this selection should be included. Even the introduction should include a discussion about gender considering how “male social gender” could be related with motivation levels.

5)      Method: What was the sampling strategy? Authors should include a discussion of the desired sample based on a power analysis, then the procedure used (i.e., who was contacted about participation), participants who were involved in the study. Description of the sample should be improved. For example,  How many time did employ to fill the instruments? Given the sensitivity of the questions how was anonymity and confidentiality conveyed and ensured? How long did the data collection process take overall?

6)      I have missed a procedure section. How were the questionnaires administered? Was required a signed consent from participants?

7)     realibility data of the scales should be included in the description of the instruments.

8)      Given that the reliability of some scales are low (.65) I suggest to calculate the omega to have a more accurate picture of the validity of the scales selected.

9)      Discussion section is a repetition of the results section. I would like to see more elaboration, for example, how do the findings relate to the theories included in the introduction?

10)   I did not find that the authors explain the implications of their results to the future of research in this area.

11)   Authors has not included a section analyzing the limitations of their research.

12) Finishing the paper I had the impression that the manuscript did not pass the “so what” test. I mean, authors should include information (at the end of the manuscript) explain how their research add to the field. As I pointed before, authors should ask themselves what implications have their research for future international research.

Author Response

Response to Reviewer 2 Comments

Point 1: Given the cross-sectional design, authors should avoid casual language. They should avoid the use of expression such as “predictort”, etc. given the study design authors should only talk about associations or relationships. 

Response 1: We were modified the use of expression such as “predictor” by "relationships" (lines 177-179 and 196-202)

Point 2: Introduction: I am not familiarize with the topic but I appreacite that authors built a strong theoretical framework to justify their study.

Response 2: Thank you very much for your comments.

Point 3: Introduction should be end with concrete hypothesis to be test. Hypothesis should be drive in previous studies. I would appreciate an inclusion of a rationale for each hypothesis tested.

Response 3: We have added the hypotheses in the introduction section (lines 94-97), and the hypothesis has been answered in the results section (lines 177-179)

Point 4: Were all the selected athletes males or females? A rationale for this selection should be included. Even the introduction should include a discussion about gender considering how “male social gender” could be related with motivation levels.

Response 4: Yes, all the athletes were men. The study was intentional, with a sample for convenience, the objective of the study does not distinguish motivational level between gender.

Point 5: Method: What was the sampling strategy? Authors should include a discussion of the desired sample based on a power analysis, then the procedure used (i.e., who was contacted about participation), participants who were involved in the study. Description of the sample should be improved. For example,  How many time did employ to fill the instruments? Given the sensitivity of the questions how was anonymity and confidentiality conveyed and ensured? How long did the data collection process take overall?

Response 5: The sample was for convenience, because participated in the study the athletes who were at the sport event. However, we have expanded the information by answering your questions, both in the section of method (lines of the procedure 129-139), and in the results (lines 158-160).

Point 6: I have missed a procedure section. How were the questionnaires administered? Was required a signed consent from participants?

Response 6: The procedure section was added (lines 129-139)

Point 7:  Realibility data of the scales should be included in the description of the instruments.

Response 7: The value of Alpha was added in the description of the instruments (Lines 102, 111,112, 122, 123)

Point 8:  Given that the reliability of some scales are low (.65) I suggest to calculate the omega to have a more accurate picture of the validity of the scales selected.

Response 8: The statistical analysis was performed, updating information in the Data Analyses section (lines 146, 147) and Results section (line 166 and a column of Table 1)

Point 9: Discussion section is a repetition of the results section. I would like to see more elaboration, for example, how do the findings relate to the theories included in the introduction?

Response 9: We add three paragraphs to answer (Lines 196 a 210)

Point 10: I did not find that the authors explain the implications of their results to the future of research in this area.

Response 10: We add two paragraphs to answer (Lines 216-229)

Point 11: Authors has not included a section analyzing the limitations of their research.

Response 11: We add one paragraph to answer (Lines 230-236)

Point 12: Finishing the paper I had the impression that the manuscript did not pass the “so what” test. I mean, authors should include information (at the end of the manuscript) explain how their research add to the field. As I pointed before, authors should ask themselves what implications have their research for future international research.

Response 12: We add one paragraph to answer (Lines 211-215)

Reviewer 3 Report

Unfortunately, the present study is not a solid evidence of the topic mentioned above. The manuscript has several major issues; however, to me the main problems are the lack of consistency among title, aim and methodology, and the ambiguous experimental design and procedure not well supported by the literature.

The introduction did not provide enough and relevant information related to the topic.  Introduction section needs to connect and support the purpose of the study based on previous evidence.

The style of the discussion section needs to be reconsidered.

The conclusions are not related to the aim of study and they are too generally.

 Author Response

Response to Reviewer 3 Comments

Point 1: Unfortunately, the present study is not a solid evidence of the topic mentioned above. The manuscript has several major issues; however, to me the main problems are the lack of consistency among title, aim and methodology, and the ambiguous experimental design and procedure not well supported by the literature.

Response 1: The article has changed substantially, the introduction has been extended, linking it with the discussion, in addition, we have added the section of the procedure.

Point 2: The introduction did not provide enough and relevant information related to the topic.  Introduction section needs to connect and support the purpose of the study based on previous evidence.

Response 2: Were added paragraphs in the introduction to better describe the theory and relationship between variables, in addition, with some cites (lines 29-30, 34-47 and 83-91)

Point 3: The style of the discussion section needs to be reconsidered.

Response 3: The article has changed substantially, the introduction has been extended, linking it with the discussion.

Point 4: The conclusions are not related to the aim of study and they are too generally.

Response 4: We have added a paragraph with the conclusion (Lines 237-241)

Round  2

Reviewer 2 Report

Authors have addressed my concerns in a satisfactory way and the study is not better described allowing replicability. I sugget to include in the title the word "male" given that the study is only focused on male volleyball players.

Author Response

The suggestion has been answered and is modified in the article.

Reviewer 3 Report

The manuscript was improved according with our recommendations. 

Author Response

Thank you very much